# Cluster Tree for Nearest Neighbor Search

**Dan Kushnir**                                                                        *dan.kushnir@nokia-bell-labs.com*
*Bell Laboratories, Murray Hill, NJ*
*NOKIA*

**Sandeep Silwal**                                                                                *silwal@cs.wisc.edu*
*Department of Computer Science*
*University of Wisconsin-Madison*

**Reviewed on OpenReview:** *https://openreview.net/forum?id=ELtNtkGXoK&noteId=sM5Bzmwuaa*

## Abstract

Tree-based algorithms are an important and widely used class of algorithms for Nearest Neighbor Search (NNS) with random partition (RP) tree being arguably the most well studied. However, in spite of possessing theoretical guarantees and strong practical performance, a major drawback of the RP tree is its lack of adaptability to the input dataset. Inspired by recent theoretical and practical works for NNS, we attempt to remedy this by introducing `ClusterTree`, a new tree based algorithm. Our approach utilizes randomness as in RP trees while adapting to the underlying cluster structure of the dataset to create well-balanced and meaningful partitions. Experimental evaluations on real world datasets demonstrate improvements over RP trees and other tree based methods for NNS while maintaining efficient construction time. In addition, we show theoretically and empirically that `ClusterTree` finds partitions which are superior to those found by RP trees in preserving the cluster structure of the input dataset.

## 1 Introduction

Nearest neighbor search (NNS) is a fundamental problem with applications in machine learning, databases, data science, and other fields and has enjoyed a vast amount of algorithmic work, both in theory and practice (see the surveys of Wang et al. (2016b); Andoni et al. (2018b); Wang et al. (2014), and Wang et al. (2016a) and references within). The problem is defined as follows: given a dataset $X \subset \mathbb{R}^d$, build a data structure over $X$ so that for future queries $q \in \mathbb{R}^d$, we can quickly return one or more datapoints in $X$ that are closest to $q$.

In this paper, we focus broadly on the space partition family of methods for nearest neighbor search. Given a query $q$, they produce a *sublinear* sized subset $P \subset X$ (referred to as the candidate set) that includes the desired neighbors. Then rather than computing distances to all points in $X$ from $q$, we instead compute a *sublinear* number of distances. In space partition methods, the subset $P$ returned is determined by space partitions that 'bucket' the points in $X$. This leads to substantially faster query times necessary for scaling to large datasets.

Space partition methods[1] have numerous advantages: they are suitable for distributed and parallel computing as different partitions can be stored on different machines (Bahmani et al., 2012; Ni et al., 2017; Li et al., 2017; Bhaskara & Wijewardena, 2018). They are also GPU friendly due to predictable memory access patterns (Johnson et al., 2021). In addition, they been used to design efficient and secure NNS algorithms (Chen et al., 2019). Lastly, they access the data $X$ in one shot, rather than multiple adaptive access, which is crucial for fast dataset construction as well as cryptographic security. Therefore, space partition algorithms are an important (and well studied) class of algorithms for NNS. See also the motivation given in Dong et al. (2020).

---

[1]many remarks apply to indexing based methods broadly of which space partitions fall under

There are two main categories of algorithms that perform space partitions: (a) tree based methods (Bentley, 1975; Uhlmann, 1991; Ciaccia et al., 1997; Katayama & Satoh, 1997; Liu et al., 2004; Beygelzimer et al., 2006; Sinha, 2015; Babenko & Lempitsky, 2017; Ram & Sinha, 2019b; Dasgupta & Sinha, 2013; 2014) and (b) hashing based methods such as Locality Sensitive Hashing (LSH) (Gionis et al., 1999; Andoni & Indyk, 2006; Datar et al., 2004; Wang et al., 2014; 2016a). Tree based methods have further advantages over hashing based methods as they are extremely fast to build (requiring roughly linear time on average), and also provide the user control over the size of sets $P$ returned for queries by setting an appropriate leaf-size. Tree based methods have been shown to outperform hashing based methods in practice as well (Sinha, 2014; Muja & Lowe, 2009; Liu et al., 2004).

The most well studied tree based algorithm is the random partition (RP) tree of Dasgupta & Sinha (2013; 2014). It uses randomness in an *oblivious* manner to recursively compute partitions of the data. Despite some theoretical guarantees and strong empirical performance of RP trees, they have a strong deficiency which motivates ours paper: **Can we utilize randomness while adapting to the underlying dataset structure for tree-based NNS algorithms?**

## 1.1 Our Contributions

We consider a new tree based method which utilizes the power of random projections as in RP trees while adapting to the underlying *cluster structure* of the dataset. We name our tree `ClusterTree`. Our contributions are as follows:

- **Fast dataset construction**: We optimize for balanced partitions (as in RP-tree) leading to fast tree construction, while also retaining other benefits of tree methods such as user level specification over the size of the returned set $P$.

- **Adapting to dataset structure**: `ClusterTree` adapts to the underlying cluster structure to find balanced partitions, by essentially modifying the split criterion of RP-trees, and selecting an optimal projection direction for the split. This leads to *meaningful and explainable partitions* which are especially important given the recent interest in explainable ML algorithms (see references within recent works such as Wan et al. (2021); Dasgupta et al. (2020) and Carvalho et al. (2019) and the workshop of XAI (2021)). Additionally, we provide hierarchical clustering analysis for `ClusterTree` in the supplemental material.

- **Theoretical analysis and Empirical advantage**: We study the performance of `ClusterTree` under natural dataset modeling assumptions and relate it to recent works on graph cuts as well as fast methods for learning Gaussian mixtures; see Sections 2.1 and 3 for more details. Furthermore, our experiments on a variety of real datasets demonstrate that our method is superior to RP trees and other tree based methods; see Section 4, and the supplemental material.

## 1.2 Related Works

We briefly overview additional algorithms for NNS besides the hashing and tree-based methods outlined in the introduction. The other class of methods besides space partitions include those where the goal is to generate compressed representations or codes of the input points so that distances can be quickly estimated (Wang et al., 2014; 2016a; Ge et al., 2014; Jégou et al., 2011; Wu et al., 2017) when a linear scan is performed (whereas we are interested in *sublinear* number of distance calculations). There have also been work to combine compressed codes with tree methods such as Product-Split trees (Babenko & Lempitsky, 2017). The fastest methods (with respect to the query time) empirically are graph based where a similarity graph is constructed over the input points (Malkov & Yashunin, 2020; Hajebi et al., 2011; Malkov et al., 2014; Wu et al., 2014). Then given a query, the graph is traversed using a greedy algorithm until convergence.

Note that space partition and tree-based algorithms, which are the focus of this paper, have several advantages over these methods. For example, the graph based search methods lack theoretical guarantees, have sub-optimal 'locality of reference' (which makes them unsuited for modern architectures (Johnson et al., 2021; Bahmani et al., 2012; Ni et al., 2017; Li et al., 2017; Bhaskara & Wijewardena, 2018; Sun et al.,

2014)), slow construction time, and require adaptive access to data; see the introduction for more benefits of tree-based methods.

We focus on tree based methods which adapt to the underlying dataset. RP trees are stated to adapt to the intrinsic dimensionality of the data and perform better for dataset possessing small intrinsic dimension (Dasgupta & Sinha, 2013; 2014). However, the RP tree construction algorithm is agnostic to structure and density and uses randomness in a data-oblivious manner. Other methods which explicitly utilize the dataset at hand include PCA trees and 2-means trees. PCA trees recursively split on the top principal component of the dataset (Sproull, 2005; Kumar et al., 2008; Abdullah et al., 2014). While more adaptive than RP trees, PCA trees can be significantly costlier to construct due to PCA computation (McCartin-Lim et al., 2012). 2-means trees on the other hand, adapt to the dataset by recursively finding partitions which minimize the 2-means cost (Dong et al., 2020). We note work on adapting the guarantees of RP trees to KD trees, but the performance of KD-trees is still worse than RP trees or PCA trees (Ram & Sinha, 2019a). Lastly, we mention that several augmentations to RP trees have been proposed, such as using sparse random projections and traversing the tree using auxiliary information (Keivani & Sinha, 2021; Hyvönen et al., 2016; Sinha & Keivani, 2017). Amongst the above tree methods, RP tree is closest to `ClusterTree` as they both utilize random one-dimensional projections. However, `ClusterTree` employs a more sophisticated algorithm to process the projections, which optimizes for balanced data partitions while adapting to the input dataset cluster structure. Lastly, we remark that one-dimensional projections have been recently used (for both theory and practice) in other settings, such as $k$-means clustering (Charikar et al., 2023).

## 2 The `ClusterTree` Algorithm

### 2.1 Motivation

In this section we motivate our algorithm for `ClusterTree`. First, we briefly outline tree based algorithms for NNS: trees are constructed starting from the root node, which represents the entire dataset. Then every node is processed by splitting the points at the node using some partition rule to create left and right child nodes. The partition rule is recursively applied to each node until each leaf node of the final tree contains at most a user specified $P$ number of points. Therefore, any tree based algorithm can be specified by its choice of partition rule. Given a query $q$, we traverse the tree, following the correct side of the partition the query lands on, until we reach a leaf node.

For RP trees, the partition rule consists of projecting points in a node to one-dimension via a random projection and then splitting based on the median (or slight perturbation of it). It's effectiveness comes from the fact that the randomness is unlikely to split a query from its true nearest neighbor.

However, picking the median split after a random projection can be sub-optimal. For example, suppose that the one-dimensional projection results in two well separated clusters where each cluster contains a non-trivial fraction of points and one cluster is slightly larger than the other. The median split passes through the larger cluster and splits it into two parts which can adversely affect the accuracy of future queries: if a query's true nearest neighbors is part of the larger cluster, we can fail to return many nearby points if we descend into the wrong part of the partition. In this case, a better choice of partition would have adapted to the cluster structure by separating the two clusters, and would have allowed for higher quality nearest neighbors to be returned. See Figure 1 for an example.

However, we still have to roughly balance every partition to ensure that the tree construction time is $\widetilde{O_d}(|X|)$. In particular for the nodes at the top level of the tree, we must ensure both parts of the partition contains a constant factor of the number of points in the node to guarantee fast construction time. To balance these two objectives, we use the well known notion of *graph conductance* which optimizes for both balanced partitions and cluster quality. Our strategy after performing a one-dimensional random projection is to form a $k$-nearest neighbor graph, for some parameter $k$, and find a conductance minimizing partition. For details, see Algorithm 1. This raises some natural questions:

- **Why graph cuts?** Graphs cuts are motivated by both recent theoretical and practical developments. On the theoretical side, there are recent works on NNS for general metric spaces that rely

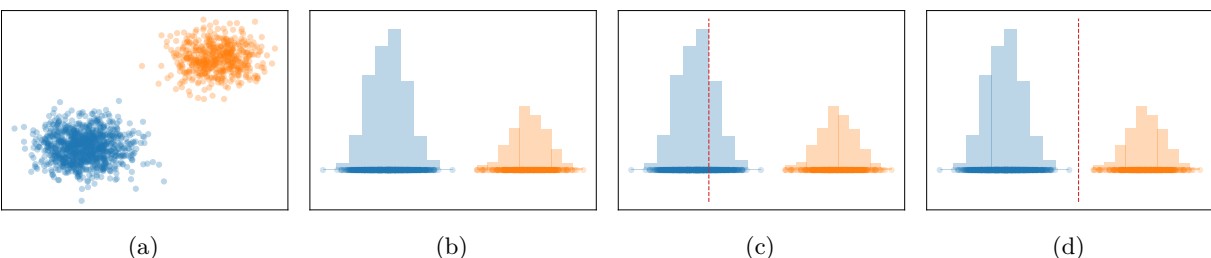

Figure 1: (a) Dataset consists of two well separated clusters. (b) Random one-dimensional projection of the dataset. Histogram denote the density of projections. (c) Partitioning strategy of RP trees which uses the median of the projection. (d) Our partitioning strategy successfully separates the clusters.

on spectral graph theory (Andoni et al., 2018c;d). On the practical side, a recent work of Dong et al. (2020) shows that learning space partitions induced from graph cuts of the $k$-nearest neighbor graph using machine learning tools leads to a very competitive algorithm for NNS. Furthermore, another popular set of algorithms for NNS is to build graphs for a dataset $X$ (such as the $k$-nearest neighbor graph) and then given a query, perform a random walk to determine the output. The intuition underlying these works is that graph structure captures properties such as clusterability, which is intimately tied to graph cuts, and is important for accurate NNS algorithms. Lastly, another advantage of graph cuts based on conductance is that it also optimizes for balanced partitions.

We note that the learning based method and the random walk based method are not in scope of this paper since both require large computational cost to build the search data structure: both require building a graph on the dataset while the learning based method further requires finding sparse cuts on the whole graph (in addition to processing it using a neural network). In addition, the second method crucially requires adaptive access to the dataset while tree based method access the data in 'one shot' which is needed for secure search such as over encrypted data (Chen et al., 2019) in addition to the multiple benefits outlined in Section 1.

- **Why one-dimensional projections?** There are practical and theoretical reasons why we perform one-dimensional projections. On the practical side, building the $k$-nearest neighbor graph in one-dimension is extremely fast (nearly linear time) as it can be computed quickly after sorting. This is *not true in larger dimensions.* Furthermore in one-dimension, there is a natural set of cuts to optimize over, which are cuts based on prefixes of the sorted order. On the theoretical side, we motivate this procedure by studying clusterable datasets under a natural Gaussian model. By relating to prior works, we show that under natural assumptions, (a) optimizing for hyperplane cuts based on prefixes leads to a 'good' partition for NNS, and (b) one-dimensional projections can capture cluster structure present in the original dimension. Lastly, we optimize over multiple random projections independently as a single projection can be very noisy; however, we can significantly increase the probability of capturing the cluster structure by trying multiple projections.

## 2.2 Algorithm

We present below our algorithm for `ClusterTree` (Algorithm 2), which employs the efficient one-dimensional cut detection described in Algorithm 1. First, we define the notion of graph conductance.

**Definition 2.1** (Conductance). *Given a graph $G = (V, E)$, $V_1 \subset V$, and $\overline{V} = V \setminus V_1$, the conductance of the cut $(V_1, \overline{V})$ is given by*

$$\varphi(V_1) = \frac{E(V_1, \overline{V})}{\min(vol(V_1), vol(\overline{V}))}$$

*where $E(V_1, \overline{V})$ is the number of edges between $V_1$ and $\overline{V}$ and $vol(S)$ denotes the sum of degrees of vertices in $S$.*

---

**Algorithm 1** OneDProjection($X, T, k$)

---

**Require:** Dataset $X \subset \mathbb{R}^d$ with $|X| = n$, $T, k \geq 0$
**Ensure:** Output partition $X = X_1 \cup X_2$, vector $v$
 1: **for** $i = 1$ to $T$ **do**
 2:     $X^i \leftarrow$ random 1 dimensional projection of $X$ using $v^i \in \mathbb{R}^d$
 3:     $Y_1, \ldots, Y_n \leftarrow$ sorted $X^i$ with each $Y_j \in \mathbb{R}$
 4:     $G^i \leftarrow k$-nearest neighbor graph on $X^i$
 5:     $\varphi_i \leftarrow \min_{1 \leq j \leq n-1} \varphi(S_j)$ where $S_j$ is the cut in $G^i$ given by $(Y_1, \ldots, Y_j), (Y_{j+1}, \ldots, Y_n)$
 6:     $w^i \leftarrow (v^i, \beta) \in \mathbb{R}^{d+1}$ is the vector encoding the projection which determines the cut
 7: **end for**
 8: **Return** the partition $X_1 \cup X_2$ induced by the cut with the smallest $\varphi_i$ value and the vector $w^i$ associated with the cut

---

The above Algorithm 1 performs our data adaptive one dimensional splitting rule. At a high level, it forms a nearest neighbor graph in one dimension after projecting the dataset, and finds the sparsest *prefix* cut in the nearest neighbor graph. $v^i$ is the random vector use to perform the random projection and $X^i$ is the resulting set of $n$ real numbers after computing the inner product with $v^i$ for every point in our dataset. We sort the one-dimensional embedding represented by $X^i$ to form $Y_1, \ldots, Y_n$ where we note that each $Y_i \in \mathbb{R}$. Finally, Algorithm 2 uses the partition strategy of Algorithm 1 to form a tree structure, and Algorithm 3 recovers the NN of an input point $q$.

---

**Algorithm 2** MakeClusterTree($X, P, T, k$)

---

**Require:** Dataset $X \subset \mathbb{R}^d$, leaf size $P, T, k \geq 0$
**Ensure:** Output Cluster Tree over $X$
 1: **if** $|X| \leq P$ **then**
 2:     **Return** leaf containing $X$
 3: **end if**
 4: $(X_1, X_2, v) \leftarrow$ OneDProjection($X, T, k$)
 5: LeftSubTree $\leftarrow$ MakeClusterTree($X_1, P, T, k$)
 6: RightSubTree $\leftarrow$ MakeClusterTree($X_2, P, T, k$)
 7: **Return** (LeftSubTree, RightSubTree)

---

**Algorithm 3** Query($q, \mathcal{T}$)

---

**Require:** Query $q \in \mathbb{R}^d$, `ClusterTree` $\mathcal{T}$
**Ensure:** Output leaf of $\mathcal{T}$ where $q$ falls in
 1: Current node $\leftarrow \mathcal{T}$
 2: **while** current node is not a leaf node **do**
 3:     Pick left or right child of current node based on its projection and bias
 4: **end while**
 5: **Return** points of $X$ in the leaf fitting the query $q$

---

**Remark 2.1.** *Each node of the tree is implicitly stores the vector $v$ used to perform the partition.*

## 3 Theoretical Analysis

We first analyze the runtime of `ClusterTree`. We start with quantifying the number of operation in OneDProjection:

**Lemma 3.1.** *The runtime of* OneDProjection *is* $O(T \cdot (nd + n \log n + nk))$.

Note that we might not be optimizing for the cut with lowest conductance since such a cut could potentially not respect the sorted ordering. However, we show in the next lemma that order preserving cuts are the sparsest cuts in the graph (fewest number of edges crossing the cut) which suggests that optimizing over prefix cuts is sufficient. We further motivate optimizing over prefix cuts with additional theoretical results in Section 3.1 and Theorem 3.5.

**Lemma 3.2.** *Consider a $k$-nearest neighbor graph on a set of $n$ points $\{X_1, \ldots, X_n\} \subset \mathbb{R}$ satisfying $X_i \leq X_{i+1}$ for all $1 \leq i \leq n - 1$. The sparsest cut respects the sorted ordering. That is, the cut with the fewest number of edges is of the form $(X_1, \ldots, X_j), (X_{j+1}, \ldots, X_n)$ for some $j$.*

We now state an assumption about the balanced partitions. Note that conductance automatically rewards balanced cuts but for worst case dataset, it can potentially find a very unbalanced cut which will lead to large tree construction.

**Assumption 3.3.** *For sufficiently large datasets $|X|$, Algorithm 1 returns a partition such that $\min(|X_1|, |X_2|) \geq c|X|$ for an independent constant $c \leq 1/2$.*

We argue that this is a valid assumption for our algorithm as Assumption 3.3 holds for datasets with very different structural properties. For example, the assumption holds for uniform inputs on one hand and also highly clustered inputs on the other hand (see Section D). In addition, we empirically verify 3.3 for real datasets as well in Section D.

**Lemma 3.4.** *Given Assumption 3.3, the tree construction time of* MakeClusterTree *is $O(T \log n \cdot (nd + n \log n + nk)) = O(Tnd \log n + Tn \log^2 n + Tnk \log n) = \widetilde{O}(nd)$ for $k, T = O(1)$ as in our experiments.*

### 3.1 Nearest Neighbor Guarantees

We now study the guarantees for `ClusterTree` for the problem of nearest neighbor search. We define two parameters, $\alpha$ and $\beta$, that have been used in the context of nearest neighbor search (Dong et al., 2020). For a given data set, $\alpha$ and $\beta$ measure the average distance squared between two $k$-nearest neighbors and the average distance squared between two arbitrary points, respectively.

**Definition 3.1.** *Let $\mathcal{D}$ be a distribution from which we sample our dataset $X$. Denote $\mathcal{D}_{close}$ to be the distribution over random $k$-nearest neighbors $(x, x') \in X$. To sample from $\mathcal{D}_{close}$, we first pick a uniformly random point $x \in X$ and then a uniformly random $k$-nearest neighbor $x'$ of $x$. Define*

$$\alpha = \mathbb{E}_{(x,x') \sim \mathcal{D}_{close}} \|x - x'\|_2^2, \quad \beta = \mathbb{E}_{x \sim \mathcal{D}, x' \sim \mathcal{D}} \|x - x'\|_2^2.$$

*Note that $\alpha$ is the expected distance squared between two 'close' points in $X$ (with respect to $k$-nearest neighbors) and $\beta$ is the expected distance squared between two random elements of $X$.*

The assumption $\alpha \ll \beta$ is natural since it states that nearest neighbors are closer than arbitrary pairs of points and thus a non-trivial algorithm is needed, rather than just returning a random point.

We utilize the following theorem from Dong et al. (2020):

**Theorem 3.5.** *There exists a hyperplane $H = \{x \in \mathbb{R}^d \mid \langle a, x \rangle = b\}$ such that the following holds. Let $X = X_1 \cup X_2$ be the partition of $X$ induced by $H : X_1 = \{x \in X \mid \langle a, x \rangle \leq b\}, X_2 = \{x \in X \mid \langle a, x \rangle > b\}$. Then, one has*

$$\frac{\Pr_{(x,x') \in \mathcal{D}_{close}}[x, x' \text{ are separated by } H]}{\min(\Pr_{x \sim \mathcal{D}}[x \in X_1], \Pr_{x \sim \mathcal{D}}[x \in X_2])} \leq \sqrt{\frac{2\alpha}{\beta}}.$$

**Remark 3.1.** *The existence of the hyperplane $H$ from Theorem 3.5 is proved using spectral graph theory and is intimately connected to a sparse cut in the $k$-nearest neighbor graph of $X$. Furthermore, Theorem 3.5 only guarantees the existence of a good hyperplane cut, rather than an arbitrary cut that may not be defined by a hyperplane. However, this is* exactly *the family of cuts we optimize for in Algorithm 1.*

Theorem 3.5 roughly states that if nearest neighbors are much closer than arbitrary pair of points, then a good hyperplane cut exists which separates the dataset into approximately balanced parts while also assuring that many $k$-nearest neighbor pairs are in the same partition. This is a natural assumption to make since otherwise, returning arbitrary points for queries could suffice for approximate nearest neighbor applications.

Note however that this is an assumption about the dataset in the *ambient* dimension, but we are finding cuts after performing a random one-dimensional projection. We argue that after such a projection, the values of $\alpha$ and $\beta$ are approximately preserved.

**Lemma 3.6.** *Suppose we sample our dataset $X$ from distribution $\mathcal{D}$. Let $P$ be a random one-dimensional projection independent of $X$ and define $PX = \{Px \mid x \in X\}$. Let $\alpha_X, \beta_X$ and $\alpha_{PX}, \beta_{PX}$ denote the values of $\alpha$ and $\beta$ for the datasets $X$ and $PX$ respectively. Then $\alpha_{PX} \leq \alpha_X$ and $\beta_{PX} = \beta_X$.*

Lemma 3.6 states that if a dataset $X$ satisfies $\alpha_X \ll \beta_X$, then the projected dataset $PX$ also satisfies $\alpha_{PX} \ll \beta_{PX}$. Then by an application of Theorem 3.5, we know that we can find a good hyperplane cut for the projected dataset. However, it is not clear that such a hyperplane would also perform well for the

original dataset with respect to nearest neighbor search since points can be heavily distorted after a random projection onto one-dimension. In the next section, we argue the soundness of performing one-dimensional projection by assuming a Gaussian mixture model. While this is a simplifying step that does not model all realistic datasets, it serves to highlight the fact that the assumption $\alpha \ll \beta$ is natural for datasets with a strong cluster structure, in addition to showing significance of trying *multiple* one-dimensional cut in Algorithm 1 and picking the best cut. This is important since a single one-dimensional cut has a high chance of returning a very 'noisy' output, even if the original dataset has a strong cluster structure. Thus trying multiple cuts boosts the probability of finding a 'good' projection.

### 3.1.1 Gaussian Mixture Model

In this section, we analyze the performance of Algorithm 1 for the mixture of two well-separated Gaussians and provide bounds on the number of projections that are needed in Algorithm 1 in terms of the mixture parameters. Our intention is to demonstrate the advantageous behaviour of `ClusterTree` in the cases of clustered data points. We start with the definition of $c$-separated Gaussians.

**Definition 3.2.** *Gaussians $\mathcal{N}(\mu_1, \Sigma_1)$ and $\mathcal{N}(\mu_2, \Sigma_2)$ in $\mathbb{R}^d$ are defined to be $c$-separated for*

$$c := \frac{\|\mu_1 - \mu_2\|}{\sqrt{d}\left(\sqrt{\lambda_1(\Sigma_1)} + \sqrt{\lambda_1(\Sigma_2)}\right)},$$

*where $\lambda_1(\Sigma)$ denotes the largest eigenvalue of the matrix $\Sigma$. We consider the case were $c$ is at least a constant value, independent of $d$.*

We first instantiate Theorem 3.5 for a mixture of two Gaussians.

**Lemma 3.7.** *Suppose that dataset $X$ with $|X| = n$ is sampled from the distribution $\mathcal{D} \sim w\mathcal{N}(\mu_1, \Sigma_1) + (1 - w)\mathcal{N}(\mu_2, \Sigma_2)$. Further, suppose that $X$ contains at least $k$ points from each of the two distributions that make up $\mathcal{D}$ and $\min(w, 1 - w) = \Omega(1)$. Define $\alpha_X$ and $\beta_X$ as in Definition 3.1. Then $\alpha_X \leq 2\max(\mathrm{tr}(\Sigma_1), \mathrm{tr}(\Sigma_2))$ and $\beta_X = \Omega(\|\mu_1 - \mu_2\|^2 + \mathrm{tr}(\Sigma_1 + \Sigma_2))$.*

**Remark 3.2.** *Note that the hypothesis in Lemma 3.7 about $X$ having at least $k$ points from each component is easily satisfied with high probability if $k = o(n)$ and $\min(w, 1 - w) = \Omega(1)$ by a Chernoff bound.*

Lemma 3.7 tells us that if $\|\mu_1 - \mu_2\|^2$ (distance between the two Gaussian means) is sufficiently large compared to $\max(\mathrm{tr}(\Sigma_1), \mathrm{tr}(\Sigma_2))$, then $\alpha_X/\beta_X$ is bounded away from 1. For example, if we have two spherical Gaussians with covariance matrices $\sigma_1^2 I_d$ and $\sigma_2^2 I_d$ respectively, then we require $d \cdot n \cdot \max(\sigma_1^2, \sigma_2^2) \lesssim \|\mu_1 - \mu_2\|^2$ for $\alpha_X/\beta_X \lesssim 1/d$ to hold. In terms of Definition 3.2, it suffices to require $c = \Omega(1)$ to guarantee $\alpha_X/\beta_X = o(1)$.

The following lemma connects the concept of $c$-separability with the guarantees of Theorem 3.5.

**Lemma 3.8.** *Suppose dataset $X$ is sampled from distribution $\mathcal{D} \sim w\mathcal{N}(\mu_1, \Sigma_1) + (1 - w)\mathcal{N}(\mu_2, \Sigma_2)$ and the conditions of Lemma 3.7 hold. Then $\alpha_X/\beta_X \lesssim 1/c^2$ for $c$ as in Definition 3.2.*

Note that the above discussion applies to the original, yet to be projected, dataset. The hope is that a well-separated pair of Gaussians will remain so after a random projection. This might not be the case as a single projection can be extremely noisy since we are projecting to an extremely small dimension. However, it is possible to derive the number of one-dimensional projections needed for well-separated mixtures to also project to well-separated one-dimensional mixtures. Thus by optimizing over *multiple* cuts in Algorithm 1, we can hope to pick a one-dimensional projection which ensures that different components remain separated after the projection.

We first need to define the $Q$ function: For $x \in \mathbb{R}$, $Q(x) = \int_x^\infty \exp(-t^2/2)\,dt$. The following result bounds the number of projections needed to achieve well-separated one-dimensional projections Kushnir et al. (2019).

**Lemma 3.9.** *Suppose our dataset $X$ is a mixture of two $c$-separated spherical Gaussians in $\mathbb{R}^d$. Let $T(c', d)$ denote the expected number of one-dimensional projections needed for the two mixtures to project to a $c'$-separated projection in one-dimension. Then we have $\lim_{d\to\infty} T(c', d) = 1/(2Q(c'/c))$.*

The following corollary states that we only need a sub-logarithmic (in $d$) number of one-dimensional projections to guarantee the same order of separation as in the ambient dimension.

**Lemma 3.10.** *If $c'$ is such that $c' \leq c(\log\log d)^{O(1)}$, then $T(c', d) = o(\log d)$.*

## 4 Experiments

In this section we evaluate our algorithm empirically on real and synthetic datasets. Our main results shows the trade-off in acceleration and accuracy of ClusterTree. As in other NNS works, we measure the fraction of actual $k$-nearest neighbors among the returned candidates (Dong et al., 2020). This metric measures the processing time required for queries since distances are computed from a query to all of the returned candidates. As seen below, and in the additional experiments we provide in our supplementary material, in the vast majority of cases ClusterTree outperforms all benchmarks in its accuracy-to-processing-time trade-off.

**Datasets.** We use the following datasets which have been used in previous machine learning works on clustering and nearest neighbor search (for example the works of Dong et al. (2020); Keivani & Sinha (2021); Lucic et al. (2018), and Bachem et al. (2018)): KDD Cup (clustering dataset from a Quantum physics task) (kdd, 2004), News (dataset of news text where each feature represents if a key word is included) (Rennie, 2016), Spam (spam text where each feature represents the presence of a particular word associated with spam) (van Rijn, 2016), SIFT (image descriptors) (Aumüller et al., 2017), and Gaussian Mixtures. See Table 1. Due to space limitation we provide experiments with additional data sets in appendix E, and F.

**Benchmarks.** Our main focus is tree-based algorithms since they are preferable in numerous settings (such as fast construction time, secure computation, and distributed and GPU architectures). Our baselines are the following. **Random Partition (RP) Trees:** This is the method from Dasgupta & Sinha (2013; 2014) and is arguably the most common tree-based nearest neighbor search algorithm. For RP trees, the partition strategy is to split along the median (or a small perturbation of the median) after performing a one dimensional random projection. **2-means Trees:** The partition strategy is to split points after performing a 2-means clustering. We use the classic $k$-means algorithm until convergence (Dong et al., 2020). **PCA Trees:** the partition strategy is to split along the median after projecting onto a principal direction (Sproull, 2005; Kumar et al., 2008; Abdullah et al., 2014). **Locality Sensitive Hashing (LSH):** While this is not a tree-based method, it is a classic space partition algorithm and the most well studied theoretical approach (see references in Section 1). We use the Cross-Polytope version from Andoni et al. (2015; 2018a).

| Dataset | $n$ (Size) | $d$ (Dimension) |
|---|---|---|
| Gaussian Mixture | $5 \cdot 10^4$ | $10^2$ |
| News | $\sim 4 \cdot 10^5$ | $10^3$ |
| RNA | $\sim 3 \cdot 10^5$ | $8$ |
| Spam | $\sim 10^6$ | $57$ |
| SIFT | $10^6$ | $128$ |
| KDD Cup | $5 \cdot 10^4$ | $78$ |

Table 1: Datasets used for our experiments.

**Parameter Selection.** In all of our experiments, we use a fixed value of $T = 20$ random projections in Algorithm 1. For the value of $k$, we initialize $k = 20$ and keep increasing $k$ by one until the value of the normalized cut found stops decreasing. Intuitively, it ensures we don't overlook a potentially better cut. Empirically, we observed that this only iterates over a few values ($\sim 5$) of $k$.

**Evaluation Metric.** As in other NNS works, we measure the number of candidates returned for queries versus the $k$-NN accuracy, which is defined to be the fraction of its actual $k$-nearest neighbors that are among the returned candidates (Dong et al., 2020). This metric measures the processing time required for queries since distances are computed from a query to all of the returned candidates. Note that tree-based methods have close to identical query costs. For example, if the trees are all approximately balanced, then on average we perform the same number of operations to return the set of candidates for queries (logarithmic number of vector operations to traverse the tree). Furthermore, the 'wall clock' time for performing queries can be heavily dependent on specific architectures and implementations. Thus, we focus on the *quality* of the partitions given by the trees. We display the average over 10 independent trials in all of our results and shade $\pm 1$ standard deviation where appropriate.

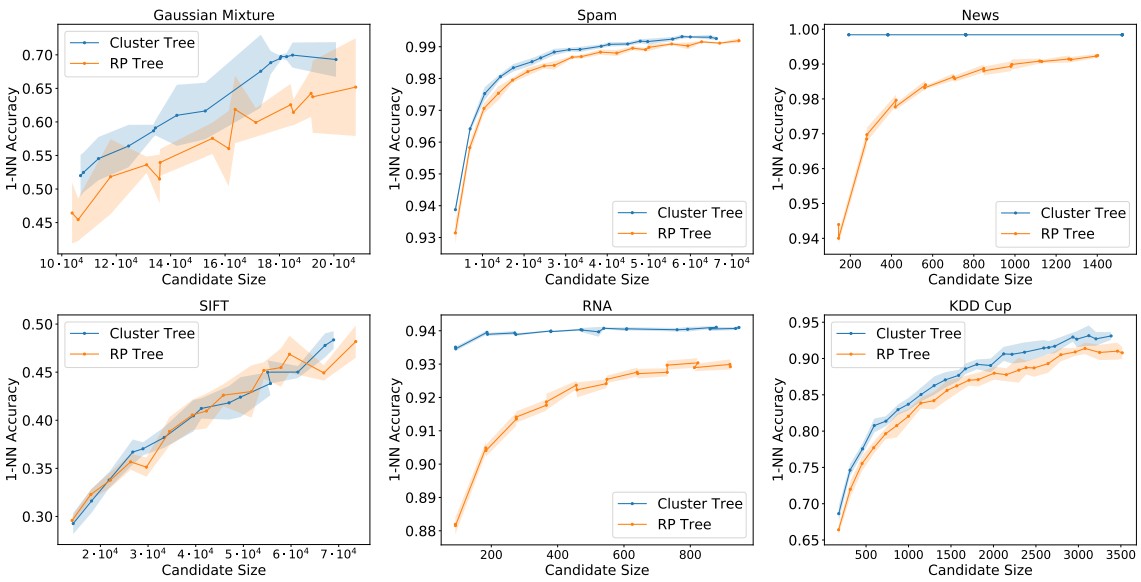

Figure 2: Candidate Size vs 1-NN Error for `ClusterTree` and RP tree for datasets in Table 1.

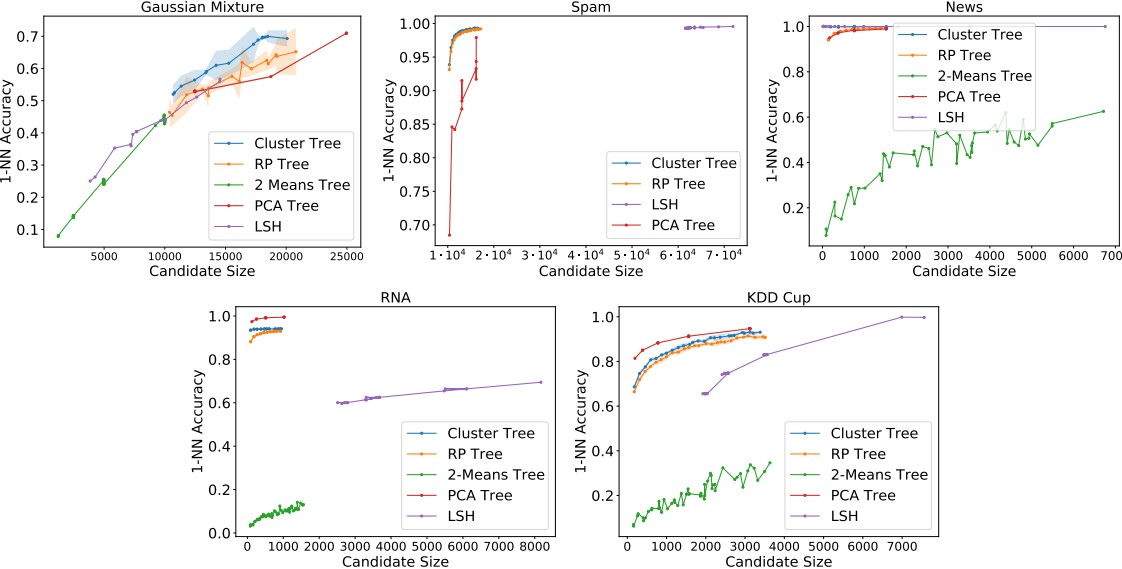

Figure 3: Candidate Size vs 1-NN Error for all baselines.

**Nearest Neighbor Experiments.** We first evaluate the performance of the algorithms on 1*st* nearest neighbor error. We ranged over various candidate sizes (by iterating over the leaf parameter) and plotted the fraction of times the nearest neighbor is in the candidate set for a query, averaged over all queries. The results for `ClusterTree` versus RP trees are displayed in Figure 2. We see that for most datasets, `ClusterTree` is *outperforming RP Trees* as fewer candidates are required to get better 1-NN accuracy. We provide results for additional six data sets in appendix E.

In Figure 3, we show the results for all of the baselines. We observe that 2-means tree performed the worst on most datasets. Note that for the two datasets KDD Cup and RNA, the computationally intensive PCA trees performed better than `ClusterTree`, while for the others: News, Spam, and Gaussian Mixtures, `ClusterTree` was the best algorithm. Nevertheless, `ClusterTree` has been consistently better than RP Trees. We also remark that it was not computationally feasible to run 2-means Tree and PCA trees for SIFT

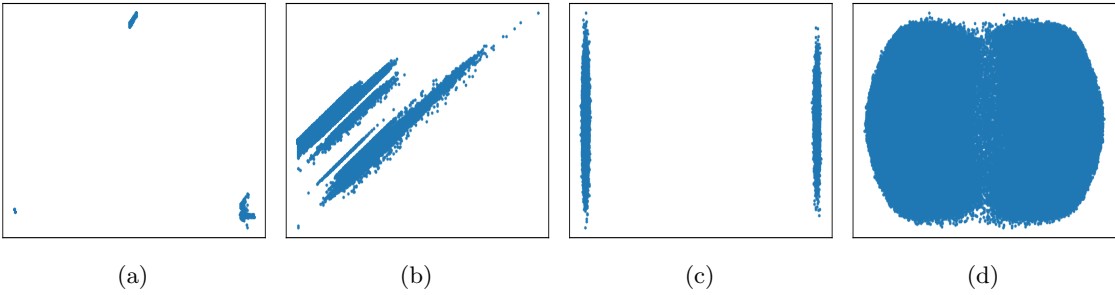

(a)              (b)              (c)              (d)

Figure 4: First two PCA axes of: (a) KDD CUP, (b) RNA, (c) GMM, and (d) SIFT.

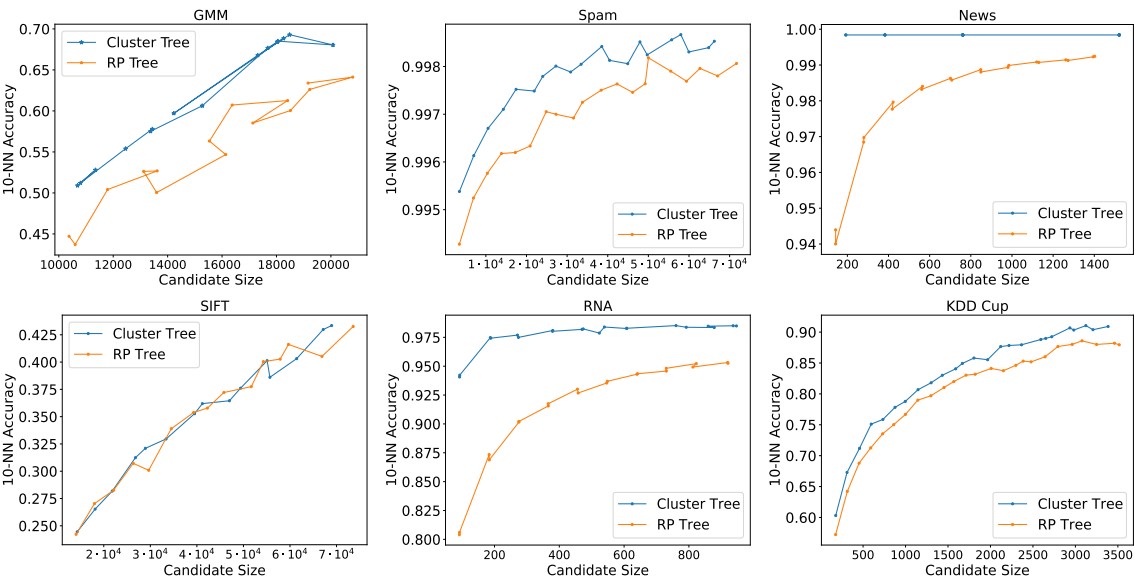

Figure 5: Candidate Size vs 10-NN Error for `ClusterTree` and RP tree.

and Spam. We are not plotting error bars for 2-means tree for clarity since it was much larger than all the other algorithms; overall, we observed 2-means trees to be an inherently unstable algorithm.

We repeat the 1-NN experiments for 10-NN in Figures 5 and 6 (also in appendix E). Note that PCA trees outperform `ClusterTree` on the KDD Cup dataset but `ClusterTree` is the best algorithm on all other datasets. Note that PCA trees and 2-means trees are costly to construct, especially for large datasets, since they are employing a much more computationally intensive partition rule than `ClusterTree` or RP trees. Lastly, LSH was not as tuneable as the tree-based algorithms in terms of specifying the approximate size of candidates to return; we could smoothly increase the candidate sizes for each tree based algorithm but LSH had a strict lower bound for the number of candidates returned per dataset, even after using a large number of hash functions, for some datasets such as Spam. This maybe due to the fact that LSH is not well suited for point sets whose norms are not well concentrated.

**Tree-Construction Running-Time.** We note that both RP Trees and ClusterTree are highly efficient to construct in practice and their main difference is in the *quality* of candidates returned on queries, for which we demonstrate `ClusterTree`'s advantage. Since we optimize over multiple projections, there is a marginal overhead of using ClusterTree over RP trees. Even on a dense dataset (SIFT) of $10^6$ point in dimension 128 with the leaf parameter set to $5 \cdot 10^3$, the runtimes to create an RP tree was 11.8 seconds on average whereas ClusterTree took 45.2 seconds. Thus, we do not envision the tree construction to be a bottleneck in practice since they only have to be constructed once.

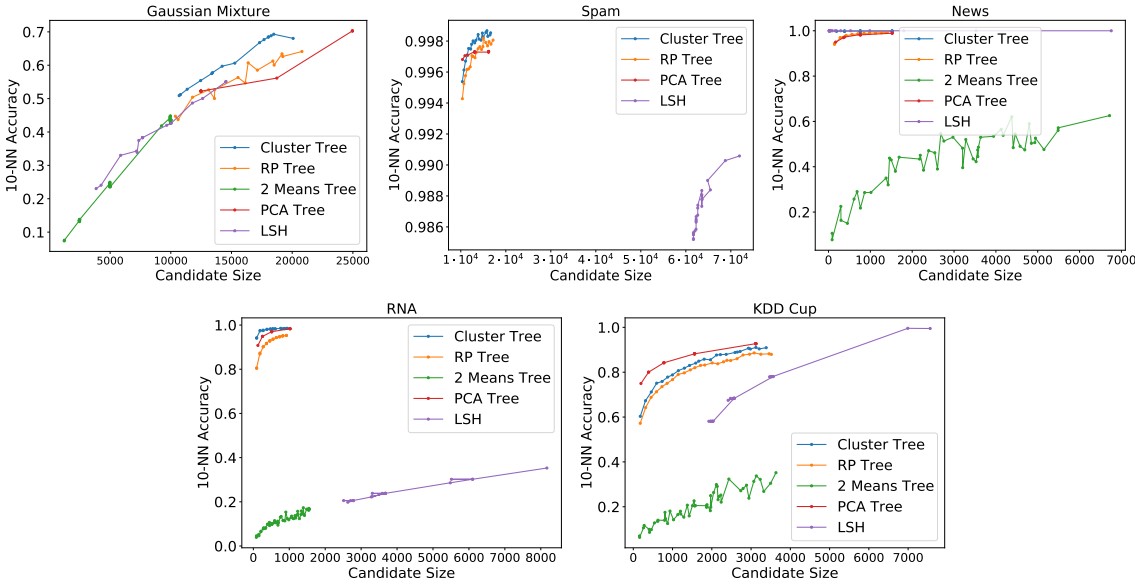

Figure 6: Candidate Size vs 10-NN Error with all baselines.

| Dataset | Accel. factor | | Bucket size | | ClTree Accuracy | | RPTree Accuracy | | Comments |
| | **min** | **max** | min | max | min | max | min | max | |
|---|---|---|---|---|---|---|---|---|---|
| GMM | ×**1.07** | ×**1.27** | 15302 | 16430 | 0.64 | 0.69 | 0.61 | 0.6 | |
| SPAM | ×**2.25** | ×**2.7** | 3489 | 24130 | 0.9980 | 0.9985 | 0.9968 | 0.9975 | |
| News | **NA** | **NA** | NA | NA | 1.0 | 1.0 | 0.95 | 0.99 | No shared y-values |
| SIFT | ×**1.0** | ×**1.07** | 14639 | 68514 | 0.35 | 0.43 | 0.35 | 0.41 | non separable (Fig. 4-d) |
| RNA | ×**6.56** | ×**7.44** | 92 | 123 | 0.94 | 0.978 | 0.95 | 0.81 | |
| KDD Cup | ×**1.34** | ×**1.48** | 2355 | 178 | 0.86 | 0.88 | 0.84 | 0.84 | |

Table 2: ClusterTree vs. RPTree query running-time acceleration min and max values for 10-NN.

**Query Running-Time of `ClusterTree` vs `RP Tree`.** The main difference between the two data structure is in the *quality* of candidates returned on queries. To this end, the experimental setting for Figures 2 and 5 can be used to calculate the acceleration in query time achieved with `ClusterTree` over RP Trees for a given accuracy, which in shown in Table 2 for the 10-NN case. The table provides the minimal and maximal acceleration factor obtained by `ClusterTree` over RP Trees for each data set. We measure the running time in terms of the number of distance computation operations used to retrieve the candidates for a given bucket size. For example, $10^4$ points in $R^{128}$, takes 0.1 seconds. Table 2 also provides the bucket sizes corresponding to the maximal and minimal acceleration obtained, and the accuracy of each method at the minimal and maximal accuracy difference between the methods. Hence, the trends observed in Table 2 largely depend on the clusterability of each data set (see discussion below and Fig. 4). We also provide similar tables for comparing ClusterTree with the two other tree-based baselines in appendix F.

**Varying Number of Projections.** We ranged over various candidate sizes and plotted the 1-NN error as the number of projections (the parameter $T$) in Algorithm 1 varied. This is to demonstrate that optimizing over *multiple* one-dimensional projections is important in practice, as suggested by our theoretical analysis. While the number of projections does not influence the performance for some datasets, we note that for datasets such as News and RNA, optimizing over multiple projections is advantageous. The results are shown in Figure 7. Note that only for this experiment, we use a randomly sub-sampled datasets Spam, News and SIFT with $5 \cdot 10^4$ points for computational efficiency.

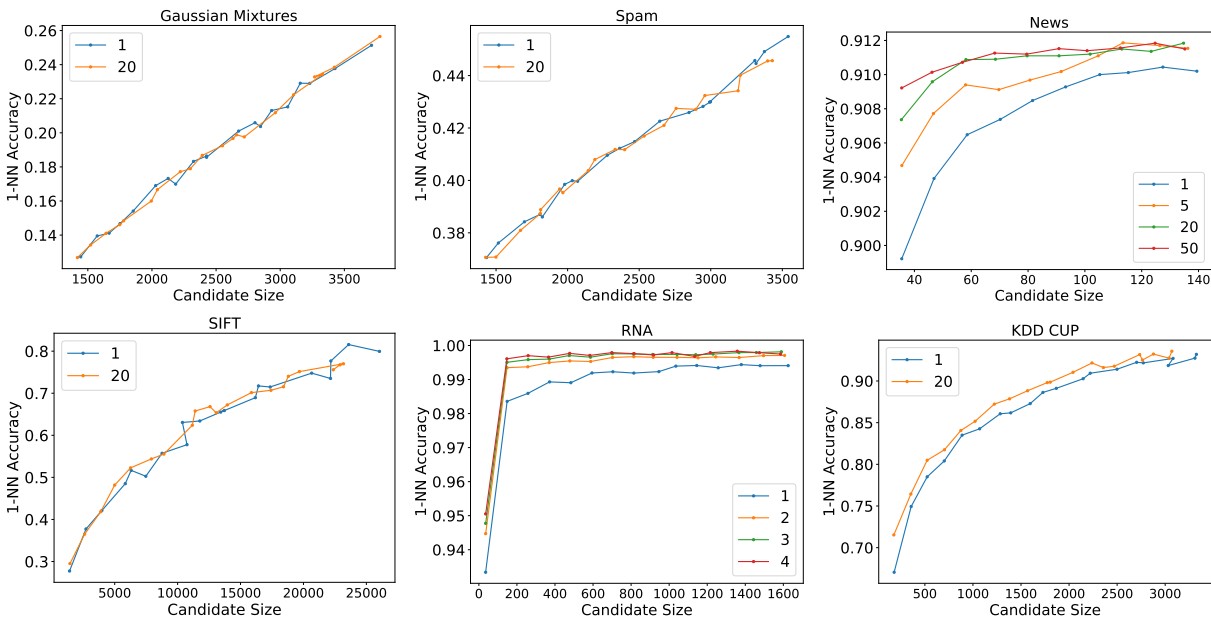

Figure 7: Optimizing over multiple projections in Algorithm 1 can improve accuracy.

**When can we expect `ClusterTree` to outperform RP trees?** Our tree construction method especially exploits the cluster structure as it builds the tree over the dataset. If the dataset does not possess such a property, we expect `ClusterTree` and RP trees to have approximately the same behaviour. To highlight this, we plotted the first two PCA projections of the centered and normalized versions of some of our datasets in Figure 4. We can observe a strong cluster structure in KDD Cup, RNA, and synthetic datasets. The projection of the SIFT dataset is mostly uniform over a region, signifying that it lacks such a structure compared to the other datasets displayed, and therefore `ClusterTree` advantage is lower, as seen above. Likewise, we can see in Figure 2 that `ClusterTree` is superior to RP trees in the 1-NN experiments for KDD Cup, RNA, and synthetic datasets whereas it is comparable to RP trees for SIFT. Our other experiments above follow a similar pattern. Therefore, we believe `ClusterTree` is preferable over RP trees as many natural datasets have a strong underlying cluster structure.

**Additional Experimental Results.** Additional experimental results are given in Appendix C, including smoothness of the tree metric, and validation that `ClusterTree` preserves cluster structure of RP-trees. We also provide more results with additional data sets in appendix E, and acceleration tables for additional tree benchmarks in appendix F.

## 5   Conclusion

We presented a novel and fast construction of a tree-based algorithm to perform fast nearest neighbor search in high dimension. Our approach utilizes randomness as in RP trees while adapting to the underlying cluster structure of the dataset to create well-balanced and meaningful partition tree. This balancedness allows a fast and accurate search of nearest neighbors. Our Theoretical analysis and the usage of the fast 1-dimensional graph-cuts provides a solid support to ClusterTree's empirical performance, in particular to its advantage over RP trees and other related benchmarks.

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

# A Omitted Proofs

## A.1 Proof of Lemma 3.1

*Proof.* Computing a single one dimensional projection takes time $O(nd)$ and computing a $k$-nearest neighbor graph can be done in time $O(nk)$ after sorting the points in $O(n \log n)$ time. Note that this *crucially* depends on the fact that we perform a one dimensional projection as nearest neighbors are determined by adjacent points on the real line. It is computationally expensive to compute such a graph in arbitrary dimensions larger than 1. Finding the sparsest prefix cut after sorting as is done in line 5 of Algorithm 1 takes linear time once the $k$-nearest neighbor graph has been constructed. Overall, the runtime of each of the $T$ procedures is $O(nd + n \log n + nk)$. □

## A.2 Proof of Lemma 3.2

*Proof.* The proof follows from the fact that if a cut does not respect the sorted ordering, then we can switch two points in opposite parts of the cut to reduce the number of edges across the cut. □

## A.3 Proof of Lemma 3.4

*Proof.* Consider the computational cost of building `ClusterTree` as a tree. We claim that at every level of the tree, we do $O(nd + n \log n + nk)$ work. To verify this, we note that $cn \log(cn) + (1 - c)n \log((1 - c)n \leq cn \log n + (1 - c)n \log n = n \log n$. Then from Assumption 3.3, there are $O(\log n)$ levels of the tree, leading to the stated runtime. (One can also use the Akra-Bazzi method to arrive at the same conclusion, see Akra & Bazzi (1998) or Leighton (1996)). □

## A.4 Proof of Lemma 3.6

*Proof.* Fix the dataset $X$. First note that for any fixed points $x, y \in X$, we have

$$\mathbb{E}\|P(x - y)\|^2 = \|x - y\|^2$$

since $P$ is independent of $X$. Now for any $x \in X$, the average distance squared from $Px$ to its $k$-nearest neighbors in $PX$ is at most the average distance squared from $Px$ to the points that were originally its $k$-nearest neighbors in $X$. This gives that $\alpha_{PX} \leq \alpha_X$. Finally, note that that the expected value of the sum of all pairwise distances after the projection is the same as the sum of all pairwise distances from our observation above. This proves $\beta_{PX} = \beta_X$, as desired. □

## A.5 Proof of Lemma 3.7

*Proof.* To prove Lemma 3.7, we will need the following auxiliary result.

**Lemma A.1.** *Suppose $x \sim \mathcal{N}(\mu_1, \Sigma_1)$ and $y \sim \mathcal{N}(\mu_2, \Sigma_2)$ Then*

$$\mathbb{E}[\|x - y\|^2] = \|\mu_1 - \mu_2\|^2 + \text{tr}(\Sigma_1) + \text{tr}(\Sigma_2).$$

*Proof.* Note that $x - y$ is distributed as $\mathcal{N}(\mu_1 - \mu_2, \Sigma_1 - \Sigma_2)$ and thus, $x - y \sim \mu_1 - \mu_2 + Az$ where $z \sim \mathcal{N}(0, I)$ and $A$ satisfies $AA^T = \Sigma_1 + \Sigma_2$. Thus,

$$\|x - y\|^2 = \|\mu_1 - \mu_2\|^2 + 2(\mu_1 - \mu_2)^T Az + z^T A^T Az.$$

Since $\mathbb{E}[z] = 0$, we have

$$\mathbb{E}[\|x - y\|^2] = \|\mu_1 - \mu_2\|^2 + \mathbb{E}[z^T A^T Az]$$

and

$$\mathbb{E}[z^T A^T Az] = \sum_{i,j} \mathbb{E}[z_i z_j](A^T A)_{ij} = \sum_i (A^T A)_i = \text{tr}(A^T A) = \text{tr}(AA^T) = \text{tr}(\Sigma_1 + \Sigma_2).$$

Putting together the above calculations gives the desired result. □

Note that we can upper bound $\alpha_X$ by the expected distance squared between two points drawn form the same component. This is because the distance to the $k$-th nearest neighbor from a fixed point will always be smaller than the distance to another point drawn from the same component (assuming our hypothesis that at least $k$ points are drawn from each component). From Lemma A.1, it follows that $\alpha_X \leq 2 \max(\operatorname{tr}(\Sigma_1), \operatorname{tr}(\Sigma_2))$.

To lower bound $\beta_X$, note that since $\min(w, 1 - w) = \Omega(1)$, the expected distance squared between two uniformly random points is at least asymptotically the expected distance squared between two points from separate components. Again using Lemma A.1, it follows that $\beta_X = \Omega(\|\mu_1 - \mu_2\|^2 + \operatorname{tr}(\Sigma_1 + \Sigma_2))$. $\qquad\square$

### A.6 Proof of Lemma 3.8

*Proof.* Lemma 3.7 tells us that

$$\frac{\alpha_X}{\beta_X} \lesssim \frac{\operatorname{tr}(\Sigma_1 + \Sigma_2)}{\|\mu_1 - \mu_2\|^2} \lesssim \frac{1}{c^2}$$

where we have used the fact that $d\,\lambda_1(\Sigma) \geq \operatorname{tr}(\Sigma)$ for a covariance matrix $\Sigma \in \mathbb{R}^{d \times d}$. $\qquad\square$

### A.7 Proof of Lemma B.1

We first need the following auxiliary results from Indyk & Naor (2007).

**Lemma A.2.** *Let $x \in S^{d-1}$ and let $P$ be a random one-dimensional Gaussian projection. Then for all $t > 0$,*

$$\Pr(|\|Px\| - 1| \geq t) \leq \exp(-t^2/8), \tag{1}$$

$$\Pr(\|Px\| \leq 1/t) \leq \frac{3}{t}. \tag{2}$$

We now proceed with the proof of Lemma B.1.

*Proof.* We first claim that the diameters of $S$ and $S'$ don't increase by a large factor after a random projection. Fix $x, y \in S$. By Eq. equation 1, the probability that $\|P(x - y)\|$ increases by a factor of $t$ is at most $\exp(-t^2/8)$. Thus for a suitable constant $c$, we have that the probability $\|P(x-y)\|$ is larger by a $c(\sqrt{\log |S|} + \log(1/\epsilon))$ factor is at most $\epsilon/(3|S|^2)$. Union bounding across all pairs in $S$ and using a similar argument for $S'$ gives us that with probability at least $1 - 2\epsilon/3$, we have that $\operatorname{diam}(PS) \lesssim \sqrt{\log |S|}\operatorname{diam}(S)$ and $\operatorname{diam}(PS') \lesssim \sqrt{\log |S'|}\operatorname{diam}(S')$.

We now claim that the sets $PS$ and $PS'$ don't come 'too' close together. Indeed, take any point $x \in S$ and $y \in S'$. We have that $\|x - y\| \geq d(S, S')$. Thus by Eq. equation 2, the probability that $\|P(x - y)\|$ shrinks by a factor of $\Omega(1/\epsilon)$ is at most $O(\epsilon)$.

Altogether, we know that with probability at least $1 - \epsilon$, all three of the following events occur:

1. $\operatorname{diam}(PS) \lesssim \sqrt{\log |S|}\operatorname{diam}(S)$,

2. $\operatorname{diam}(PS') \lesssim \sqrt{\log |S'|}\operatorname{diam}(S')$,

3. $d(PS, PS') \geq \epsilon\, d(S, S') - \operatorname{diam}(PS) - \operatorname{diam}(PS')$.

Thus by our assumption that $S$ and $S'$ are $r$-apart for the value of $r$ in the lemma statement, it follows that

$$\operatorname{diam}(PS) \lesssim \sqrt{\log |S|}\operatorname{diam}(S) - \operatorname{diam}(PS) - \operatorname{diam}(PS') \lesssim \epsilon\, d(S, S') \lesssim d(PS, PS')$$

and a similar statement holds for $S'$, proving the lemma. $\qquad\square$

## A.8   Proof of Lemma B.2

*Proof.* The proof follows from Lemma B.1 as every point in $PS$ will be closer to any other point in $PS$ than any other point in $PS'$. A similar symmetric statement holds for $PS'$. Thus, any edges of the $k$-nearest neighbor graph starting from any point in $PS$ must have its other vertex in $PS$ as well. This implies that there is an empty cut between $PS$ and $PS'$, as desired. □

**Lemma A.3** (Follows from corollary 5 and 6 in Kushnir et al. (2019))**.** *Consider two c-separated Gaussian distributions in $\mathbb{R}^d$ with means $\mu_1, \mu_2$ and covariance matrices $\Sigma_1$ and $\Sigma_2$. Define $T(c', d)$ as in Lemma 3.9. Let $\gamma := 2d(c')^2 \lambda_{\max} / \|\mu_1 - \mu_2\|^2$, where $\lambda_{\max}$ denotes the largest eigenvalue of the matrix $\Sigma_1 + \Sigma_2$. Then*

$$\lim_{d \to \infty} T(c', d) \leq \frac{1}{2Q(\sqrt{\gamma})}.$$

# B   Hierarchical Clustering

In this section, we discuss the performance of `ClusterTree` for hierarchical clustering. Since our tree is designed to preserve the underlying cluster structure of the dataset, it is very natural to use it for clustering applications, such as hierarchical clustering. In hierarchical clustering, the goal is to design a tree over the input dataset which hopefully captures 'multi-scale' clustering relationships of the dataset.

To formalize this, we first define a natural hierarchical clustering model and then prove results which suggest that `ClusterTree` is naturally suited to recover such a clustering. Note that traditional algorithms for hierarchical clustering, such as computing the minimum spanning tree, require $\Omega(n^2)$ time, which is prohibitive for large datasets, whereas `ClusterTee` construction is nearly linear time.

**Definition B.1** (Hierarchical Clustering Model)**.** *Let $X$ be our dataset and $P$ be a parameter. We assume there is a tree $\mathcal{T}$ over $X$ such that the following is satisfied:*

- *The leaves of $\mathcal{T}$ are disjoint subsets of $X$ of size at most some parameter $P$ and together include all points of $X$,*

- *Level $i \geq 1$ of $\mathcal{T}$ is a union of two subsets in level $i - 1$ of the tree where level $0$ denotes the leaves. We assume that each subset at level $i - 1$ contributes to exactly one subset in level $i$ of the tree*

- *The largest level of the tree is the entire dataset $X$.*

Note that the above definition naturally describes a hierarchical clustering model over the dataset $X$ where going up the tree indicates larger scale cluster structure over the dataset $X$. We further assume a separability criteria for our hierarchical clustering model.

**Definition B.2.** *Let $\operatorname{diam}(S)$ denote the diameter of the subset $S \subseteq X$ and $d(S, S')$ denote the distance between two subsets $S, S'$:*

$$d(S, S') = \min_{x \in S, y \in S'} \|x - y\|.$$

*We say that subsets $S, S'$ are $r$-**apart** if*

$$Cr \max(\operatorname{diam}(S), \operatorname{diam}S') \leq d(S, S')$$

*for some constant $C$.*

If we assume the above definition applies to a pair of subsets of the tree $\mathcal{T}$ at any some fixed level, then intuitively we are requiring the two subsets constitute well separated clusters.

Given such an assumption, we want to argue that repeated application of Algorithm 1 can successfully recover the underlying tree $\mathcal{T}$. The intuition behind this is that if the subsets are projected, they will also be separated after a random projection with high probability. The following lemma shows that it is indeed the case.

**Lemma B.1.** *Suppose subsets $S$ and $S'$ satisfy Definition B.2 with $r \geq \sqrt{\log(|S| + |S'|)}/\epsilon$. Let $PS$ and $PS'$ respectively denote a random one-dimensional projection of the two subsets. Then with probability at least $1 - \epsilon$, we have*

$$c \max(\text{diam}(PS), \text{diam}(PS')) \leq d(PS, PS')$$

*for some constant $c > 1$.*

Lemma B.1 hints that with a sufficient separability assumption, the $k$-nearest neighbor graph in one-dimension will mostly have edges within a given cluster which leads to sparse cuts between different subsets. Thus, we can reasonably expect Algorithm 1 to separate the distinct clusters in $\mathcal{T}$ since it optimizes for sparse cuts. Formally, we can prove the following statement.

**Lemma B.2.** *Let $S$ and $S'$ be two $r$-apart subsets of dataset $X$ for the value of $r$ in Lemma B.1 and $P$ be a random one-dimensional projection. If $\min(|S|, |S'|) \geq k$, then the $k$ nearest neighbor graph of $PS \cup PS'$ will have a cut that separates $PS$ and $PS'$ with probability $1 - \epsilon$.*

Now consider the hierarchical clustering model given in Definition B.1 and let $\mathcal{T}$ denote the implicit tree over a dataset $X$. Consider a node of $v$ of $\mathcal{T}$ and at some level $i$ let $S$ and $S'$ denote the subsets at level $i - 1$ that comprise $v$. If $S$ and $S'$ are $r$-apart for a sufficiently large value of $r$, then Lemma B.2 states that $S$ and $S'$ will have an empty cut between them after a random one-dimensional projection. If we further assume that each of the two pieces of the $k$-nearest neighbor graph is connected, Algorithm 1 will exactly split apart $S$ and $S'$, as intended in the tree $\mathcal{T}$.

**Preserving Cluster Structure of the Dataset.** We empirically validate the hypothesis that `ClusterTree` is superior to RP trees in finding partitions that preserve the underlying cluster structure of the dataset. We designed two related experiments to demonstrate this. For the first set of experiments, we measured the diameter of the leaves (weighted by the leaf sizes) of each class of trees as the parameter $P$ increases. Again the intuition here is that if the diameter of the leaves are small, then it mostly contains points that are well-clustered together while conversely, if the diameter is large, then the tree has bucketed together points that belong to different clusters. Our results are shown in Figure 8. Indeed, we see that for most datasets `ClusterTree` results in leaves that are much more tightly clustered than RP Trees, which again demonstrates that `ClusterTree` is adaptive to the underlying cluster structure of the dataset.

We present additional experiments on how the weighted radius of leaves of various tree-based algorithms varies as a function size in Figure 9. See Section 4 for more details on experimental setting. Overall, we see that `ClusterTree` has a smaller radius as a function of cluster size for the Gaussian Mixture, Spam, RNA, and KDD Cup datasets. Note that for Spam, 2-means tree was too costly to run and for Gaussian Mixture, the 2-means tree and `ClusterTree` have very identical curves for weighted radius as a function of candidate size.

## C Omitted Experimental Results

We give additional experimental results in this section.

**Distance to $k$-th Nearest Neighbors.** We created instances of `ClusterTree` and RP trees for all of our datasets where we set the leaf size, the parameter $P$ in Algorithm 2, to be equal to 10% of $n$. We then computed the distance from a query to the $k$-th nearest neighbor among the candidates returned by a tree for various values of $k$ and averaged this across all queries. The intuition here is that if a leaf node of a tree contains points from multiple *distinct* clusters, then there will be a substantial increase in this metric at some intermediate value of $k$. Indeed, this is what we observe in Figure 10. For example in the Gaussian Mixture, KDD Cup, and Spam datasets, there is a noticeable 'jump' in the plots for RP trees as it is 'mixing' multiple clusters in the leaf nodes while for `ClusterTree`, the relationship is much smoother.

To recap, the intuition here is that if a leaf node of a tree contains points from multiple *distinct* clusters, then there will be a substantial increase in this metric at some intermediate value of $k$. For example, suppose that the leaf of a node contains points from two distinct well-separated clusters and consider a query that

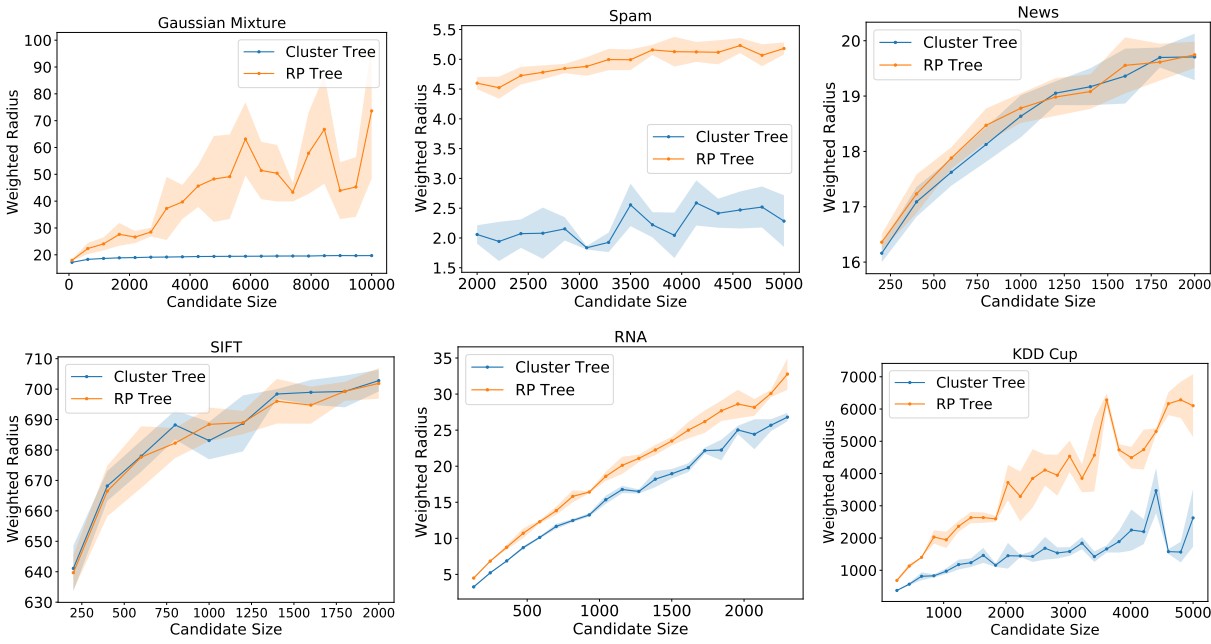

Figure 8: The leaves of `ClusterTree` have a smaller diameter than those of RP Trees.

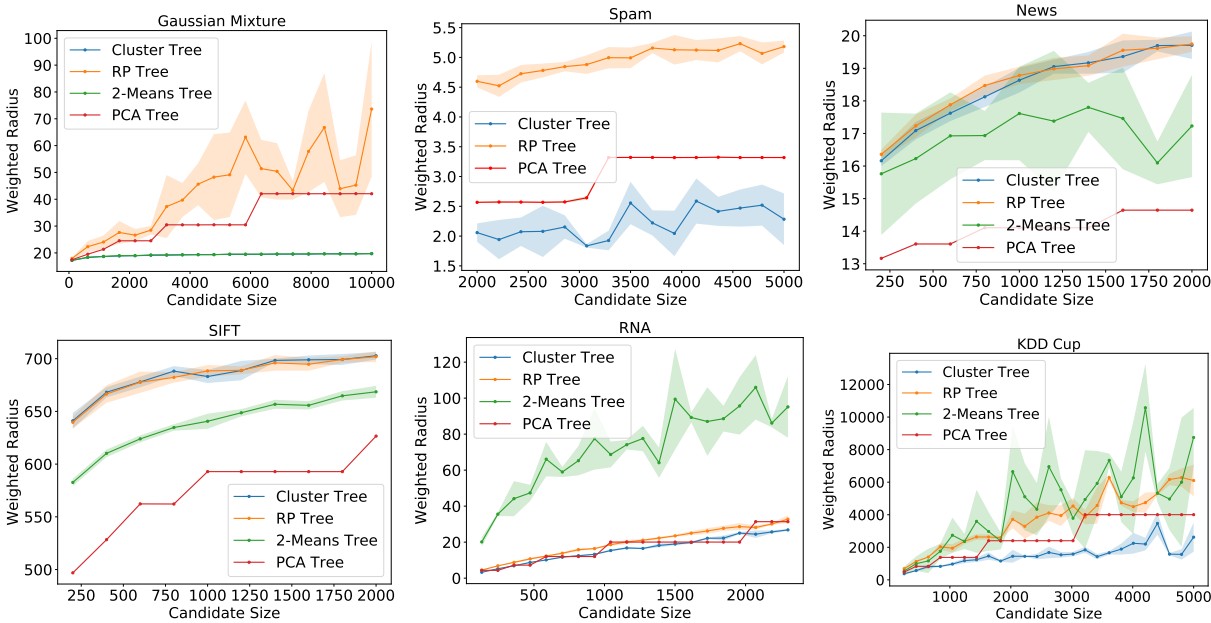

Figure 9: Expanded version of Figure 8 using all tree-based algorithms.

lands in this leaf but is closer to only one of the clusters. Then for a large enough value of $k$, the $k$-th nearest neighbor for this query would come from the far away cluster, leading to a significant increase in the distance to the $k$-th nearest neighbor in comparison to the $(k-1)$-th nearest neighbor. In contrast, if the leaf mostly contained points from one cluster, the distance would smoothly increase. To summarize, this is exactly the behaviour observed in Figure 10: in the Gaussian Mixture, KDD Cup, and Spam datasets, there is a noticeable 'jump' in the plots for RP trees as it is 'mixing' multiple clusters in the leaf nodes while for `ClusterTree`, the relationship is much smoother.

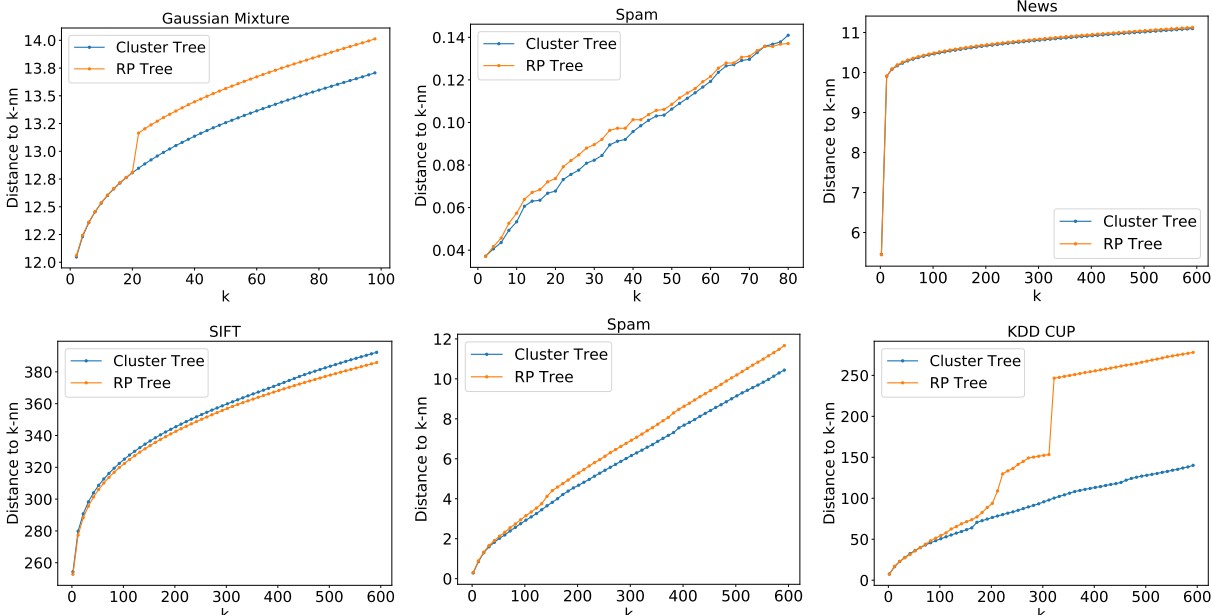

Figure 10: `ClusterTree` has a smoother trade-off curve for distance to the $k$-th neighbor as $k$ increases.

**Additional Parameter Selection Details.** If there are multiple cuts that have conductance 0, i.e., multiple separated pieces in the $k$-nearest neighbor graph constructed in Algorithm 1, we pick the cut that is the most balanced. This is because any choice of the cuts would have been good with respect to preserving near neighbors so we should optimize for keeping the tree balanced.

Note that there have been recent works on improving RP trees using additional techniques such as sparse random projections (Sinha & Keivani, 2017), using auxiliary information when performing search over the tree (Keivani & Sinha, 2021), and other methods (Hyvönen et al., 2016). For simplicity, we did not use these techniques as they can be used identically for `ClusterTree` as for RP trees.

Finally, we highlight that both RP tree and `ClusterTree` are randomized algorithms. Therefore, they have the following additional benefit: in order to boost accuracy, we can initialize multiple instances of the data structure to create an ensemble of trees while keeping the overall number of candidates fixed. For example, instead of creating one tree with leaf size $P$, we can create two trees with leaf sizes $P/2$ each. In general, if we make a constant number of trees, this can be thought of as significantly boosting accuracy by increasing the amount of space used by only a constant factor. Note that 2-means trees and PCA trees are deterministic so they do not have this additional benefit. For simplicity however, we only compare single instantiation of each algorithm.

## D Justifications for Assumption 3.3

We provide justification for Assumption 3.3 as its conditions hold true for one-dimensional datasets with very different structural properties: both uniform and clustered inputs.

- **Uniform Points**: Suppose the input to Algorithm 1 is a set of uniformly spaced points in one-dimensions. Then it is clear that the cut with the lowest conductance will split the dataset exactly in half due to symmetry.

- **Clustered Points**: Suppose the input consists of two well-separated clusters, with each cluster consisting of at least $c$-fraction of the total input size. Then the $k$-nearest neighbor graph for this input will be such that the edges of each cluster will be mostly to other points of the same cluster.

Thus, the cut separating the two clusters will be extremely sparse and hence have low conductance as well. See Figure 1 for an example.

**Average Split Ratio.** We now empirically validate Assumption 3.3. To do so, we compute `ClusterTree` for all of our datasets setting $P = 5\%$ of the size of the dataset in each case. The results across one run of Algorithm 2 are shown in Table 3. We observe that on average, each node of the tree splits the dataset into two approximately balanced parts.

| Dataset | Avg. Split Ratio | Dataset | Avg. Split Ratio |
|---------|------------------|---------|------------------|
| KDD Cup | 0.49(0.26) | Spam | 0.50(0.27) |
| News | 0.50(0.00) | SIFT | 0.49(0.18) |
| RNA | 0.45(0.29) | Gaussian Mixture | 0.53(0.12) |

Table 3: The average split ratio across all nodes in the tree with standard deviation in the parenthesis using 5% of the number of points as the parameter $P$ (leaf size).

## E  Additional Datasets

We provide additional experiments for comparison between RP-trees and ClusterTree on a set of additional data sets: two clustering benchmark data sets from (Fränti & Sieranoja, 2018), 8776x13, 10126x15, Gas sensor data set 120kx128 (Fonollosa), stocks 94701x6 (Kaggle, 2023), mediamill 43907x120 (Repository), skyserver (Ahumada et al., 2020) 500Kx11.

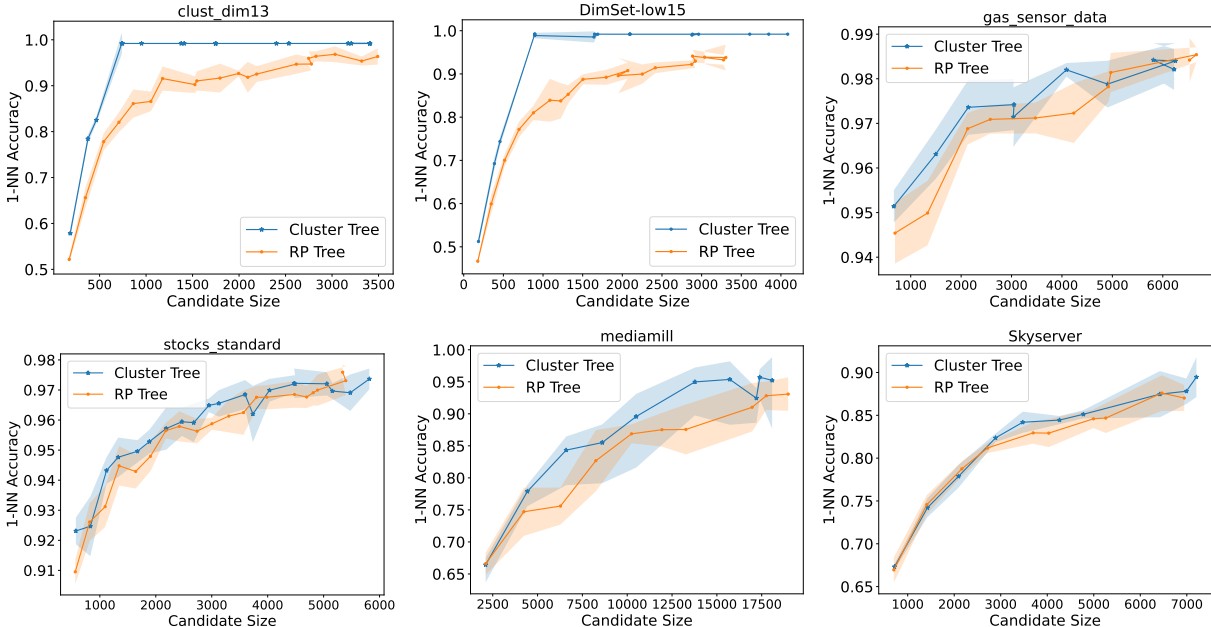

Figure 11: Candidate Size vs 1-NN Error for `ClusterTree` and RP tree.

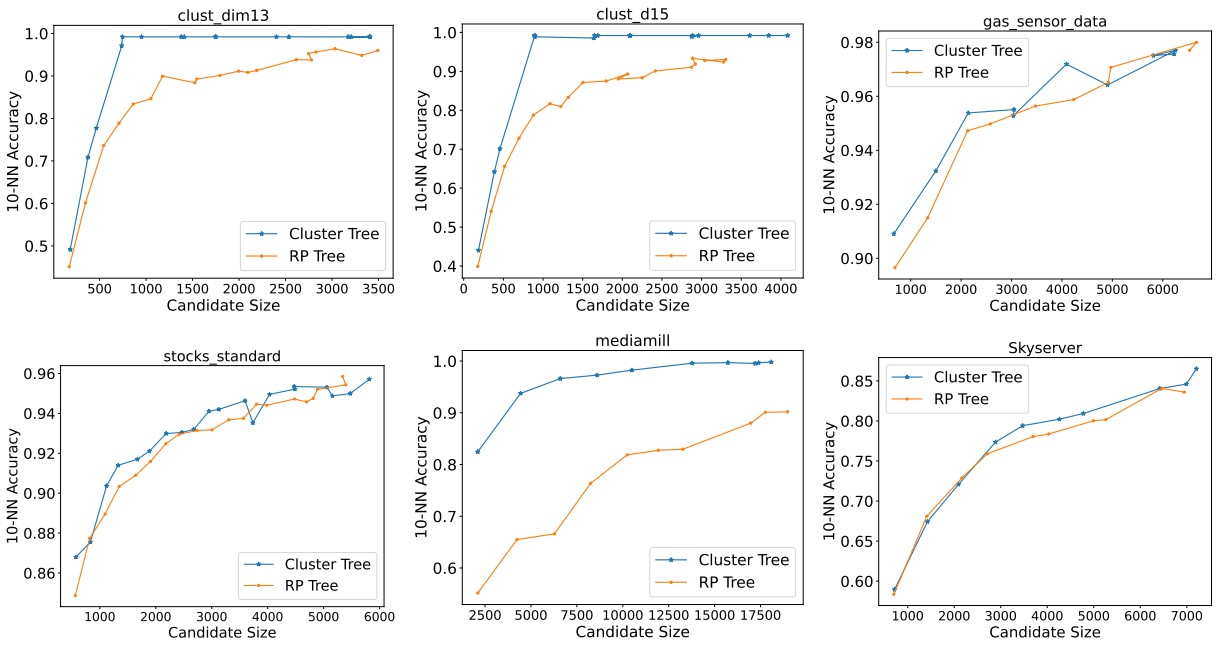

Figure 12: Candidate Size vs 10-NN Error for `ClusterTree` and RP tree.

## F  Additional Running-Time Accleration Analysis

| Dataset | Accel. factor | | CLTree bucket size | | ClTree Acc. diff | | PCATree Acc. diff | | Comments |
|---------|------|------|------|------|------|------|------|------|----------|
|         | **min** | **max** | min | max | min | max | min | max | |
| GMM | ×**1.12** | ×**1.35** | 11180 | 17676 | 0.55 | 0.69 | 0.52 | 0.56 | |
| SPAM | ×**1.15** | ×**2.7** | $3 \times 10^4$ | $5 \times 10^4$ | 0.9980 | 0.9985 | 0.9968 | 0.9975 | |
| News | **NA** | **NA** | NA | NA | 1.0 | 1.0 | 0.9 | 0.95 | no shared y values |
| RNA | **2.45** | **2.49** | 379 | 92 | 0.99 | 0.99 | 0.12 | 0.17 | |
| KDD Cup | × **0.53** | ×**0.33** | 1965 | 200 | 0.90 | 0.61 | 0.57 | 0.93 | |

Table 4: ClusterTree vs. PCATree query running-time acceleration min and max values for 10-NN.

| Dataset | Accel. factor | | CLTree bucket size | | ClTree Acc. diff | | KMsTree Acc. diff | | Comments |
|---------|------|------|------|------|------|------|------|------|----------|
|         | **min** | **max** | min | max | min | max | min | max | |
| GMM | **NA** | **NA** | NA | NA | NA | NA | NA | NA | no shared y and x values |
| SPAM | ×**1.15** | ×**2.7** | $3 \times 10^4$ | $5 \times 10^4$ | 0.9980 | 0.9985 | 0.9968 | 0.9975 | |
| News | **NA** | **NA** | NA | NA | 1.0 | 1.0 | 0.44 | 0.15 | no shared y values |
| RNA | **NA** | **NA** | NA | NA | 0.98 | 0.96 | 0.15 | 0.05 | no shared y values |
| KDD Cup | **NA** | **NA** | NA | NA | 0.60 | 0.86 | 0.07 | 0.18 | no shared y values |

Table 5: ClusterTree vs. KMEANsTree query running-time acceleration min and max values for 10-NN.

