# OpenReview forum: "Cluster Tree for Nearest Neighbor Search"
_TMLR — Accepted by TMLR_

### Review · Reviewer_ZSdM · 2024-11-28

**Summary Of Contributions:**

This paper introduces ClusterTree, a novel tree-based algorithm for nearest neighbor search that combines random projections with adaptation to dataset structure. The key innovation is modifying the traditional Random Partition tree approach by incorporating cluster-aware partitioning based on graph conductance minimization in 1D projected space. The authors demonstrate both theoretical guarantees and practical improvements over existing methods.

**Audience:**

Yes

**Claims And Evidence:**

Yes

**Requested Changes:**

Here are my suggestions in addition to the weakness section (critical issues):

1. Make better use of the \cite and \citep. Some references are put in the wrong format.
2. Section 4 doesn't seem to adhere to the same topic. You might wanna consider (a) move it to appendix (b) explain why that matters -> add some experiments for that as well.

Overall I think the paper is interesting and offers a novel solution to the nearest neighbor search problem.

**Strengths And Weaknesses:**

**Strengths**

1. Well-written paper, I enjoyed reading it. The authors laid out the main ideas nicely.
2. Through theoretical analysis. The authors provided theoretical analysis for (a) runtime complexity (b) guarantees on nearest neighbours (c) analysis on hierarchical clustering
3. The paper is accompanied with empirical results that demonstrates stronger performance.

**Weakness**
1. The assumptions are restricted. Data distribution has to be a mixture of 2 Gaussians.
2. Lack of a runtime flop analysis. The exact compute overhead is still unclear.
3. On that point, a similar table can be provided for run-time complexities of the ClusterTree and the baselines.
4. Potential missing cases where there are multiple such clusters, would ClusterTree still outperform random partition/PCA trees?

---

> ### Author Response · Authors · 2024-12-16
> **Response to reviewer ZSdM**
>
> We thank the reviewer for his excellent remarks that helped us improve the manuscript, and for his overall positive view of our work.
>
> **Weakness**
> 1. **The assumptions are restricted. Data distribution has to be a mixture of 2 Gaussians.**
> ClusterTree is not restricted to a mixture of 2 Gaussians. In our paper we prove guarantees for a mixture of 2 Gaussians. Since the algorithm can be applied iteratively to create a tree, clearly the data can be for a mixture of >2 Gaussians. As for other data distributions, this is yet to be proven, we chose to prove our algorithms guarantees for a mixture of Gaussians because this is the most ubiquitous model for data distribution.
> The hierarchical clustering section (now in the appendix) explains and analyzes why ClusterTree can work well for multi-cluster data sets, and provides the necessary guarantees and an experiment showing the improved tightness of clusters recovered with ClusterTree vs RP-trees.
> 2. **Lack of a runtime flop analysis. The exact compute overhead is still unclear.**
> We provide in the revision a table that compares the compute overhead trade-off with accuracy for the two algorithms.
> Table 2 shows the acceleration in compute overhead for the 10-NN case. The table provides the minimal and maximal acceleration factor obtained by ClusterTree over RP-Tree for each data set. We measure the running time in terms of the number of distance computation operations used to retrieve the candidates for a given bucket size, where distance running time computation is fixed.
> **On that point, a similar table can be provided for run-time complexities of the ClusterTree and the baselines**.
> We have now added this table as well to the revision
> 3. **Potential missing cases where there are multiple such clusters, would ClusterTree still outperform random partition/PCA trees?**
> In Fig 6 ClusterTree outperforms the random partition and PCA tree on the RNA data set which is shown in Fig 4 to be composed of multiple clusters (shown using a visualization of the first PCA projections).
>
> **Requested Changes:**
> **Here are my suggestions in addition to the weakness section (critical issues):**
> 1. **Make better use of the \cite and \citep. Some references are put in the wrong format.**
> We went through the paper, we are not sure where such problems occur can you please provide an example, and we will be happy to fix it. **This point is now fixed.**
> 2. **Section 4 doesn't seem to adhere to the same topic. You might wanna consider (a) move it to appendix (b) explain why that matters -> add some experiments for that as well.**
> We moved section 4 to the appendix, and relevant experiments are there.

---

> > ### Comment · Reviewer_2srD · 2024-12-18
> >
> > The issue with your citations is that the whole citation, not only the year, should be in parentheses, unless the citation used as a part of the sentence. So, for instance, "See also the motivation given by Dong et al. (2020)" is correct, but "In addition, they been used to design efficient and secure NNS algorithms Chen et al. (2019)" should be "In addition, they been used to design efficient and secure NNS algorithms (Chen et al. 2019)". This is really easy to fix, just use \citep-command.

---

> > > ### Author Response · Authors · 2024-12-19
> > > **Thank you for the clarification**
> > >
> > > Thank you for the clarification. We have changed the citation in the updated revision.

---

> > > > ### Comment · Reviewer_2srD · 2024-12-20
> > > >
> > > > Now the citations where the author names are used as a part of the sentence, and that were originally in the right format, are now wrong. As I stated in my last comment, "See also the motivation given by Dong et al. (2020)" was correct, but you have now changed it to incorrect "See also the motivation given by (Dong et al. 2020)". See, e.g, https://www.jmlr.org/format/format.html for instructions on how to format citations correctly.
> > > >
> > > > You have also fixed the format of the citations only until the beginning of Section 4. This is a very elementary formatting issue, and you should spend at least some effort to get it right, especially after it has been pointed out by Reviewers.

---

> > > > > ### Author Response · Authors · 2024-12-20
> > > > > **Response to Reviewer 2srD**
> > > > >
> > > > > Dear Reviewer 2srD,
> > > > >
> > > > > We sincerely apologize for the mistake. There was a small miscommunication on our part and we believe the citations have now been appropriately changed to use \cite{} vs \citep{}.
> > > > >
> > > > > Many thanks,
> > > > > The authors.

---

### Review · Reviewer_2srD · 2024-12-04

**Summary Of Contributions:**

The article considers the task approximate nearest neighbor (ANN) search, specifically partition-based methods for ANN search. The article improves the widely applied random projection (RP) tree data structure by making it more adapted to the cluster structure of the data. In particular, the authors propose a novel split criterion RP trees. Instead of splitting at the median of the projected node points, the authors propose building a one-dimensional $k$-nn graph of the node points, and splitting so that graph conductance, i.e., the normalized (by the smaller of the sums of the degrees of the nodes of two groups resulting from the split) number of cut edges is minimized. In addition, the authors propose a technique of trying multiple randomly chosen projection directions, and selecting one whose best split (selected via the method described above) minimizes the graph conductance.

The experimental results demonstrate that when defeatist search (a point is selected into the candidate set if and only if it belongs to the same leaf as the query point) is used to select the candidate set, on the data sets that have a cluster structure the proposed method outperforms the original RP tree that uses the median split. The authors also provide a theoretical guarantee in the special case where the data is from the well-separated Gaussian mixture model.

**Audience:**

Yes

**Broader Impact Concerns:**

I do not find any unaddressed ethical concerns with the paper.

**Claims And Evidence:**

Yes

**Requested Changes:**

I have two requests that I describe in more detail above in Strengths and Weaknesses section. These are critical for securing my recommendation:
* Outline the contribution of the article as a new split criterion for RP trees instead of a new type of data structure.
* Perform the ablation experiment described above.

**Strengths And Weaknesses:**

Strengths: The task that the article considers (ANN search) is important. The proposed method seems to improve performance of RP trees on the clustered data sets. The technical quality of the article is good. Theoretical guarantees are provided.

Weaknesses: I disagree with the way the contribution of the article is formulated. The authors state as their contribution that they introduce a novel data structure called ClusterTree. However, ClusterTree is just a RP tree that uses a different split criterion instead of the median (or a uniformly random point from the midquartile range) of the projections (and tries multiple projection directions). As I outline above in Summary of Contributions, a clearer way of formulating the contribution of the article would be to state that it proposes a novel split criterion for RP trees. I do not think that every slight variation of an existing and established data structure should be named as a new data structure.

Specifically, hyperplane trees (see, e.g., Hyvönen 2023, p.15–18), such as RP trees, PCA trees, and $k$-d trees, first project the node points onto a line, and then split the node points by a hyperplane that is orthogonal to this line. Thus, there are two components in which the hyperplane trees can differ from each other: (1) a _projection vector_ $z \in \mathbb{R}^d$ that defines the line onto which the node points are projected and thus the orientation of the splitting hyperplane; (2) a _split criterion_ that defines the location of the splitting hyperplane given the projected node points.

The main novelty of the article on the second of these components (in addition, a method for selecting the projection vector among multiple candidates is proposed). In addition to being conceptually clearer, outlining the contribution as a novel split criterion would enable applying it immediately to the other hyperplane trees, such as $k$-d trees or PCA trees. Also there is very little systematic work studying split criteria of space partition trees, so this would make the article a more valuable contribution to literature.

The article is missing ablation experiments. Two related techniques for improving RP trees are proposed: (a) selecting a split point as a graph cut that minimizes graph conductance; (b) selecting the projection vector (among randomly generated candidates) whose optimal cut minimizes graph conductance. Experiments should be performed to show what is the performance improvement (over the original RP trees) of (a) alone; and (a) and (b) combined.

The results of the article are far away from the current state-of-the-art for approximate nearest neighbor search (see e.g., Aumüller et al., 2020 or https://github.com/erikbern/ann-benchmarks for a performance comparison of the current SOTA methods). However, improving tree-based ANN search may be of interest to at least some of TMLR's audience, so I do not see an issue with that.

Martin Aumüller, Erik Bernhardsson, and Alexander Faithfull. ANN-benchmarks: A benchmarking
tool for approximate nearest neighbor algorithms. _Information Systems_, 87:101374, 2020.

Ville Hyvönen: _A machine learning approach to approximate nearest neighbor search_. PhD dissertation, University of Helsinki, 2023.

---

> ### Author Response · Authors · 2024-12-16
>
> We thank the reviewer for his important comments and overall positive view of our experimental and theoretical contribution. We have answered the two requests that the reviewer made and updated our revision to refine the outline of our contribution and address the ablation study.
>
> **Requested Changes:**
>
> **I have two requests that I describe in more detail above in Strengths and Weaknesses section. These are critical for securing my recommendation:**
> 1. **Outline the contribution of the article as a new split criterion for RP trees instead of a new type of data structure.**
> We edited the text to outline the contribution as a new tree split criterion. We note, however, that the efficient selection of the projection vector is also a part of our novelty here, and that does not exist in the RP-tree construction. We have updated the text to reflect the feedback from the reviewer, in section 1.1 bullet 2.
> 2. **Perform the ablation experiment described above.**
> We would like to direct the reviewer to the experiment reported in our supplemental material in appendix B, and now is in the experiment section as 'Varying Number of Projections' and Fig 7. This experiment shows the selection of a single random projection – as in option (a) proposed by the reviewer. And, it is compared with option (b) for varying number of selected projections.  To make sure this point is not overlooked we  moved it from the supplemental material into the body of the paper in the updated revision.

---

> > ### Author Response · Authors · 2025-01-20
> > **Final revision is posted addressing all comments and changes reqeusted.**
> >
> > We thank you for your helpful comments and suggestions.
> > In addition to our above responses we addressed the reviewer's point on running time analysis with RP tree and with the other tree baselines. The first table is provided in the experiments section, and the other tables are provided in appendix F. This concludes addressing all the weak points and the requests for change that have been raised.

---

> > > ### Comment · Reviewer_2srD · 2025-01-22
> > >
> > > Thanks for the addressing my concerns and providing additional experimental results. I do not have any further concerns, and I have updated my recommendation.

---

### Review · Reviewer_K1CX · 2024-12-07

**Summary Of Contributions:**

The paper provides a comprehensive exploration of tree-based algorithms for Nearest Neighbor Search (NNS), focusing on the limitations of random partition (RP) trees, one of the most widely studied methods in this area. While RP trees are valued for their theoretical guarantees and practical performance, they lack adaptability to the structure of input datasets. To address this, the authors propose ClusterTree, a novel tree-based algorithm that combines the randomness of RP trees with a clustering-aware approach to create well-balanced and meaningful partitions. Through both theoretical analysis and empirical evaluations on real-world datasets, the paper demonstrates that ClusterTree outperforms RP trees and other tree-based methods, offering superior accuracy and better preservation of the underlying cluster structure.

**Audience:**

Yes

**Claims And Evidence:**

Yes

**Requested Changes:**

The main concern has to do with the weak experimental evaluation. The paper mainly focus in this subarea. Unclear where the method stands (and this area in general) vs. strong quantization and graph-based solution. PQ outperforms most hash-based solutions, so someone wonders if outside of the nice theoretical contributions, these methods are not competitive in practice.

More datasets should also be used. Modern studies present results in dozens of datasets

Li, Wen, et al. "Approximate nearest neighbor search on high dimensional data—experiments, analyses, and improvement." IEEE Transactions on Knowledge and Data Engineering 32.8 (2019): 1475-1488.

**Strengths And Weaknesses:**

Strengths

- Simple yet effective algorithm to build a tree structure based on clustering
- Algorithm offers strong theoretical guarantees
- Experimental results support such theoretical claims in real world settings

Weaknesses

- The similarity search area has heavily been explored. The work mainly focuses on a narrow sub-area.
- It's unclear how competitive is this sub-area vs. new product quantization variants, graph-based solutions, etc.
- Missing baselines outside of the sub-area
- Missing datasets (there are dozens of datasets)

---

> ### Author Response · Authors · 2024-12-18
> **Response to Reviewer K1CX**
>
> Thank you for your valuable feedback.
>
> >  Unclear where the method stands (and this area in general) vs. strong quantization and graph-based solution.
>
> We agree that in practice other method such as graph-based algorithms or product quantization have much better overall performance if we only care about query time versus accuracy. However, the focus of our work is not on general graph-based approaches or PQ. Rather, we focus on the subarea of **tree-based nearest neighbor search algorithms.** There are several reasons for this:
>
> 1) Tree based algorithms can be theoretically analyzed unlike general graph constructions, for example our work or the classic RP trees.
> 2) Tree based methods as they are extremely fast to build (requiring roughly linear time on average), compared to general graph-based algorithms. This maybe advantages in situations where index construction time is also a bottleneck.
> 3) Tree based algorithms give users a finer control over the size of sets returned for queries by setting an appropriate leaf-size.
> 4) Tree based methods maybe better for certain distributed or GPU architectures (see the intro for references).
>
> However, maybe most importantly, we view our work as contributing to the growing body of work on the analysis of *data-dependent* nearest neighbor search algorithms. A fact that the reviewer is also well-aware of is that the best empirical nearest neighbor search algorithms are extremely data-dependent: the index they built is heavily over-fitted to the underlying dataset to be searched. Such data-dependency is extremely hard to theoretically analyze and we hope our work introduces a fresh tools in analyzing such algorithms. In our work, we show that underlying cluster structure of a dataset can be exploited both empirically and theoretically. Thus we view our work as among the first steps towards theoretical analysis of general data-dependent nearest neighbor algorithms such as graph-based algorithms.
>
> > More datasets should also be used
>
> In addition to the 6 datasets we extensively studied in our submission, we have included 10-nearest neighbor experiments for two other datasets in the appendix of the updated paper. Lastly, we also would like to highlight our other results that the reviewer may have missed, including some theoretical guarantees for hierarchical clustering in appendix B and additional experiments in appendix C.

---

> > ### Author Response · Authors · 2025-01-20
> > **update to revision**
> >
> > Thank you again for your helpful review.
> > In addition to the above response, we added to our final revision six additional data sets for which we report accuracy vs running time tradeoff for ClusterTree and RPtree in appendix E. In total we have now a dozen different data sets that we covered. We hope this will address your comment on sufficient experimental validation.

---

> > > ### Comment · Reviewer_K1CX · 2025-02-05
> > >
> > > Thanks for addressing my comments. I understand the theoretical focus of this work and I am glad to see more experimental evidence in the revised form.

---

### Author Response · Authors · 2024-12-20
**Extension**

Dear Reviewers,

Thank you for your useful comments and suggestions to improve the manuscript. We have addressed most of your comments, yet, we have been given by the editor in chief an extension of time to complete two tasks. The current extension is until January 20th 2025.

Thank you for your patience and happy holidays!

The authors

---

### Decision · Action_Editor_YXwx · 2025-02-06

**Recommendation:** Accept as is

**Comment:**

The paper makes a clear technical contribution for a well defined problem, and detailed theoretical and empirical evaluation is provided for validating the claims. The reviewers found the contribution somewhat incremental and straightforward, but overall it is a solid contribution for the field that meets the TMLR evaluation criteria. All comments on writing and technical content were addressed during the discussion.

**Audience:**

Two distinct audiences will find the paper interesting. The core audience, people specifically working on efficient nearest neighbor search, is small but clearly part of the ML community and for them the paper is valuable with detailed theoretical analysis. In addition, the empirical results are useful for anyone looking to use these methods in concrete applications. This audience is much broader, given the prevalence of nearest neighbor search in broad range of ML methods.

**Claims And Evidence:**

The paper proposes a new method for fast discovery of nearest neighbors by leveraging cluster structure in the data, validating the method with both extensive theoretical analysis (incl. computational cost, backing up claims on efficiency) and range of empirical experiments where it is compared against previous methods using standard benchmark data sets and metrics. The main claim, of improved performance over random trees in cases where the underlying data has clear cluster structure, is concretely demonstrated.